# Impact of Sequential Lipid Meals on Lymphatic Lipid Absorption and Transport in Rats

**DOI:** 10.3390/genes13020277

**Published:** 2022-01-30

**Authors:** Qi Zhu, Qing Yang, Ling Shen, Jie Qu, Meifeng Xu, David Q.-H. Wang, Patrick Tso, Min Liu

**Affiliations:** 1Department of Pathology and Laboratory Medicine, University of Cincinnati College of Medicine, Cincinnati, OH 45237, USA; zhuqu@ucmail.uc.edu (Q.Z.); yangqa@ucmail.uc.edu (Q.Y.); shenln@ucmail.uc.edu (L.S.); quje@ucmail.uc.edu (J.Q.); xume@ucmail.uc.edu (M.X.); tsopp@ucmail.uc.edu (P.T.); 2Department of Medicine and Genetics, Albert Einstein College of Medicine, Bronx, NY 10461, USA; david.wang@einsteinmed.edu

**Keywords:** two-meal feeding, mucosal mast cells, chylomicron, lymph fistula model

## Abstract

The sequential meal pattern has recently received more attention because it reflects a phycological diet style for human beings. The present study investigated the effects of the second lipid meal on lymphatic lipid absorption and transport in adult rats following a previous lipid meal. Using the well-established lymph fistula model, we found that the second lipid meal significantly increased the lymphatic output of triglycerides, cholesterol, phospholipids, and non-esterified fatty acids compared with a single lipid meal. Besides that, the time reaching the peak of each lipid output was significantly faster compared with the first lipid meal. Additionally, the second lipid meal significantly increased the lymphatic output of apolipoprotein A-IV (ApoA-IV), but not apolipoprotein B-48 (ApoB-48) or apolipoprotein A-I (ApoA-I). Interestingly, the triglyceride/apoB-48 ratio was significantly increased after the second lipid meal, indicating the increased chylomicron size in the lymph. Finally, the second lipid meal increased the lymphatic output of rat mucosal mast cell protease II (RMCPII). No change was found in the expression of genes related to the permeability of lymphatic lacteals, including *vascular endothelial growth factor-A* (*Vegfa*), *vascular endothelial growth factor receptor 1* (*Flt1*), *and Neuropilin1* (*Nrp1*). Collectively, the second lipid meal led to the rapid appearance of bigger-sized chylomicrons in the lymph. It also increased the lymphatic output of various lipids and apoA-IV, and mucosal mast cell activity in the intestine.

## 1. Introduction

Elevated postprandial blood triglyceride (TG) concentrations, which are mainly caused by an increase in chylomicron (CM) production, are now considered a causal risk factor for low-grade inflammation and atherosclerotic cardiovascular disease [1,2]. The sequential meal pattern has received more attention, since previous studies have mainly focused on the single-meal effect on nutrient absorption and related metabolism, which is physiologically not reflective of the scenario in human feeding. Given that sequential meal intake is a typical eating pattern in Western countries, it is important to examine how a two-meal pattern affects the absorption and lymphatic transport of lipids [3]. 

Dietary fat (mainly in the form of TG) is hydrolyzed into fatty acids (FAs), glycerol, and monoglycerides by pancreatic lipase in the intestinal lumen [4]. These digestive products are taken up by the enterocytes, where the majority of re-esterified TG is packaged to assemble CMs and secreted into circulation via the lymphatic system. A certain amount of lipids may also be stored in the enterocytes and form cytoplasmic lipid droplets (CLDs). Previous findings have shown that there is a rapid appearance of CMs in circulation following sequential fat meals. These CMs carry fat components from the first meal [5,6,7]. These findings suggest that a certain level of fat from the first meal remains in the gut and enters circulation via the stimulation derived from subsequent lipid meal ingestion. Previous studies have also revealed that glucose from the second meal stimulates the formation and mobilization of CLD in the small intestine [8]. However, those studies mainly focused on lipoproteins in the plasma, but not directly from the lymph due to technical difficulty and the risks involved in human trials. It remains unclear how the lipid output in the lymph is changed after the second lipid meal. Therefore, the study of the lymph would reveal the direct absorption and secretion of lipids from the small intestine. 

Intestinal mucosal mast cells (MMCs), which reside in the lamina propria underneath the epithelium, are important sentinel cells that sense foreign substances entering the mucosa. Upon activation, MMCs are degranulated and release cytokines, histamine, proteoglycans, and proteases. Our previous study found that dietary lipids induce the activation of MMCs in the small intestine, which may be related to low-grade inflammation in the gut [9]. However, the effect induced by the second lipid meal on the activity of MMCs is still unknown. 

We hypothesize that the second lipid meal could increase the lipid output in the lymph and further enhance MMC activation to a higher level, leading to altered intestinal function. To test this hypothesis, we used the conscious rat lymph fistula model to examine the lymph flow rate, lymphatic lipid output, apolipoprotein, and MMC-related protein expression. Furthermore, we analyzed the expression of genes that are related to lymph lacteal function. 

## 2. Materials and Methods

### 2.1. Animals

Male adult Sprague Dawley rats (300–350 g) were purchased from Envigo (Indianapolis, IN, USA) and acclimated in the animal facility at the University of Cincinnati Medical Center for 2 weeks before the experiments. The animals were allowed ad libitum access to food (Harlan-Teklad LM485 chow, Madison, WI, USA) and water prior to the experiment’s onset. All animal procedures were performed in accordance with approval by the University of Cincinnati Internal Animal Care and Use Committee and in compliance with the National Institutes of Health Guide for the Care and Use of Laboratory Animals. 

### 2.2. Two Lipid Meal Study

The rats received surgery for implanting lymph and gastric cannula, as we conducted previously [10]. Briefly, after an overnight fast, the rats were anesthetized with isoflurane and a midline incision was made. The superior mesenteric lymphatic duct was cannulated with polyvinylchloride (PVC) tubing. A gastric infusion tube was placed into the stomach. Following the surgery, the animals were infused via the gastric tube with a saline solution containing 5% glucose at 3 mL/h overnight to compensate for fluid and electrolyte loss due to lymphatic drainage. The animals were allowed to recover overnight in Bollman cages, which were kept warm in a temperature-regulated chamber (~30 °C). Analgesics were provided to alleviate pain during the experimental period. 

After recovery overnight, the fasting lymph was collected for 1 h before lipid infusion (that is, the fasting lymph) while the rats were infused with the 5% glucose/saline solution. The animals were randomly divided into two groups: one group received two meals and one group received one meal. The rats in the two-meal group received a bolus infusion of 3 mL 20% Intralipid (Sigma-Aldrich, St. Louis, MO, USA) via the stomach tube as the first meal. The intralipid composition was 20% soybean oil, 1.2% egg yolk phospholipids, 2.25% glycerin, and water, with 2.0 kcal per ml. Then, the liquid infusion was stopped to prevent stomach overexpansion. Thirty minutes later, the animals were continuously infused with a saline solution containing 5% glucose at 3 mL/h. After 6 h from the first meal, all rats received the same infusion of 3 mL Intralipid via the gastric tube, which was exactly the same process as described in the first lipid meal infusion. Lymph samples were collected hourly during the entire 12 h experimental period. At the end, i.e., 6 h after the second meal, all rats were sacrificed and the jejunum was collected. Each lymph sample contained 10% by volume of an antiproteolytic cocktail (0.25 M EDTA, 0.80 mg/mL aprotinin, and 80 U/mL heparin). 

### 2.3. Measurement of TG, Cholesterol (CHOL), Phospholipids (PL), and Non-Esterified Fatty Acids (NEFA)

The TG content was determined with Total TG assay kits (Randox Laboratories, Kearneysville, WV, USA) and CHOL using Infinity CHOL Liquid Stable Reagent (Fisher Diagnostics, Thermo Scientific, Middletown, VA, USA). The PL content was measured using Phospholipids C reagent (Wako Diagnostics, Mountain View, CA, USA) and NEFA using HR series NEFA-HR (2) Reagent (FUJIFILM Medical Systems USA, Miami, FL, USA). All assays were performed according to the manufacturers’ instructions.

### 2.4. Measurements of Apolipoproteins and RMCPII 

Apolipoproteins A-I, A-IV, and B-48, and rat mucosal mast cell protease II (RMCPII) were quantitated by Western blotting. Lymph samples (2 μL) were loaded onto 4–15% polyacrylamide gradient gels (Bio-Rad Laboratories, Inc.) and transferred to polyvinylidene difluoride membranes. Following transfer, the membranes were blocked with 5% nonfat milk in 0.1% Tween 20 in Tris-buffered saline (TBS) for 1 h. The membranes were incubated with antibodies raised against apoA-I (1:5000), A-IV (1:5000), B-48 (1:8000), and RMCPII (each 1:1000) overnight. After rinsing with TBS with 0.1% Tween 20, the membranes were incubated with peroxidase-conjugated anti-goat secondary antibody at 1:10,000 for 30 min. The bands were visualized with Immobilon Western Chemiluminescent HRP substrate (EMD Millipore, Billerica, MA, USA). Images from the reacted membranes were acquired and the band density was analyzed by ChemiDoc Imaging Systems (Bio-Rad Laboratories, Inc.). The fold-changes in lymphatic apolipoprotein output, including apolipoprotein A-I (apoA-I), apolipoprotein A-IV (apoA-IV), apolipoprotein B-48 (apoB-48), and rat mucosal mast cell protease II (RMCPII), during each subsequent hour of infusion were quantified by dividing the relative lymph apolipoprotein level at each hourly time point by the 0 h (fasting) level.

### 2.5. Quantitative Real-Time PCR (qPCR) for the Expression of Target Genes 

The total RNA was extracted from the jejunum using a Qiagen RNA extraction kit (Qiagen, Germantown, MD, USA). RNA was quantified using a NanoDrop 2000 (Thermo FisherScientific, Waltham, MA, USA). Then, 200 ng of the total RNA from each sample was used for reverse transcription to cDNA with a Transcriptor First Strand cDNA Synthesis Kit (ThermoFisher Scientific, Waltham, MA, USA). The mRNA levels of vascular endothelial growth factor-A (*Vegfa*), vascular endothelial growth factor receptor 1 (*Flt1*), and Neuropilin1 (*Nrp1*) were quantified by qPCR using TaqMan Fast Advanced Master Mix with TaqMan Gene Expression Assays with a StepOneTM Plus device (ThermoFisher Scientific, Waltham, MA, USA). The β-actin mRNA levels of each sample were used as internal controls to normalize the mRNA levels. 

### 2.6. Statistical Analysis and Sample Size

Lipid output was calculated as the product of lymph flow and each lipid’s (TG, CHOL, PL, and NEFA) concentration. Similarly, the relative protein output was calculated as the product of lymph flow and each protein abundance. The data are shown as mean values ± standard error of mean (SEM). For comparison of the data that have 2 independent variables, two-way repeated-measures analysis of variance (ANOVA) and the Tukey test were used. Statistical analyses were performed using GraphPad Prism V8. *p* values less than 0.05 were considered statistically significant. 

## 3. Results

### 3.1. Lymph Flow Rate 

As shown in Figure 1, the average fasting lymph flow rates for the two groups of rats varied from 2.3 mL/h to 2.8 mL/h. In the two-meal group, the lymph flow rates increased and reached a maximum output between 3.0 and 4.1 mL/h at 2 and 8 h after the infusion of each lipid infusion, respectively. The lymph flow rates then declined slightly and maintained a steady-state output of ~3 mL/h during the whole process after lipid infusion. In the one-meal group, the lymph flow rates remained constant during 1–6 h and then increased, reaching a maximum output of 3.5 mL/h at 9 h and maintaining a steady output afterward. There was no significant difference in the lymph flow rates between the two groups at any time point in the experimental period, indicating that the two-meal treatment did not significantly affect the lymph flow rate in the lymph fistula rats. 

### 3.2. Lymphatic TG Output 

When the rats in the two-meal group received the first dose of intralipid, the TG output was increased significantly during 2–4 h, compared with those in the one-meal group (no lipid infusion at this period), and this was consistent with our previous reports [11,12]. It is interesting to find that the TG output was significantly increased after the second meal during 7–9 h compared with the one-meal group (*p* < 0.05) (Figure 2a). Since the levels of TG output in the two-meal group were still higher at 6 h, when the second meal was infused, we decided to deduct the levels of TG output at 6 h to eliminate the impact from the previous meal (i.e., the sixth lymph TG output was considered as the fasting lymph output to the two lipid-meal animals). We found that the net increase in TG output was still significantly higher in the two-meal group at 8 h (Figure 2b) compared with that in the one-meal group, but the TG outputs at the other time points were comparable between the two groups. 

### 3.3. Lymphatic CHOL Output 

A similar trend was found in the CHOL output in the lymph. After the rats in the two-meal group received the first dose of intralipid, the CHOL output was increased, but the difference was not significant between the two-meal and the one-meal groups. It is interesting to find that the CHOL output significantly increased after the second meal at 7 h compared with that in the one-meal group (*p* < 0.05) (Figure 3a). However, the net CHOL output was comparable between the two groups (Figure 3b). 

### 3.4. Lymphatic PL Output 

When the rats in the two-meal group received the first dose of intralipid, the PL output was significantly increased during the period of 2 h to 4 h compared with the PL output at the same time points in the one-meal group. In addition, the PL output was significantly increased after the second meal during 7 to 9 h compared with that in the one-meal group (*p* < 0.05) (Figure 4a). The net PL output (Figure 4b) was also significantly higher than that in the one-meal group at 8 h, and the area under the curve from the time period of 7 to 12 h in the two-meal group was also significantly increased compared to that in the one-meal group (25.7 ± 2.30 vs. 15.3 ± 3.52, *p* < 0.05).

### 3.5. Lymphatic NEFA Output

The lymphatic NEFA output was comparable between the two groups during the first dose of intralipid infusion (Figure 5a). Interestingly, when the rats in the two-meal group received the second dose of intralipid, the NEFA output significantly increased at 8 h in the two-meal group compared with that in the one-meal group (*p* < 0.05) (Figure 5b). 

### 3.6. Lymphatic Output of Apolipoproteins and RMCPII

We next examined the secretion of apolipoproteins (apo), including apoA-I, apoA-IV, apoB-48, and RMCPII in the lymph of the two groups of rats. As shown in Figure 6a,c, the hourly lymphatic outputs of apoB-48 and apoA-I were comparable between these two groups of rats. Interestingly, the second lipid meal significantly increased the apoA-IV levels in the lymph at 7 h and 8 h, and the RMCPII levels at 8 h (*p* < 0.05) (Figure 6b,d).

### 3.7. Chylomicron (CM) Size Analysis

As a surrogate measure of CM particle size, we took advantage of the fact that each CM only contained one apoB48. Thus, the ratio of lymph TG to apoB-48 is reflective of the size of the CM and was calculated at each time point. A larger ratio indicates a larger particle size. At 4 h and 8 h, the two-meal group had a significantly higher ratio of TG to apoB-48 (*p* < 0.05) (Figure 7), indicating that a larger CM size appeared after the second lipid meal. 

### 3.8. Expression of Genes Regulating Intestinal Lacteal Structure and Function

In order to study the molecular mechanisms underlying the alterations of lipid profiles induced by two lipid meals, we measured the mRNA levels of three important genes that regulate lacteal structure and function, including vascular endothelial growth factor-A (*Vegfa*), vascular endothelial growth factor receptor 1 (*Flt1*), and Neuropilin1 (*Nrp1*), in the rat jejunum. Although there were trends to increase the levels of *Vegfa* and *Flt1* mRNAs in the two-meal rats, the difference did not reach a statistically significant level (Figure 8). 

## 4. Discussion

A two-meal pattern protocol was used in the present study to mimic the eating patterns of normal individuals [3,13]. The second-meal effect has been reported previously that the intestine has a substantial capacity to store fat [5,6,13]. In this study, we aimed to assess whether the second lipid meal further increased intestinal lipid output in the conscious lymph fistula model of rats. We found that there were significantly increased levels of various lipids (TG, CHOL, PL, and NEFA) and apolipoprotein output (apoA-IV) in the lymph from the rats receiving two lipid meals compared to those animals that received one single lipid meal with an extended 6 h fasting time. Although apoB-48 output after the second lipid meal infusion was not significantly changed, the rats on two lipid meals had increased TG to apoB-48 ratios, indicating increased particle sizes of chylomicrons (CMs). Therefore, while the number of CMs secreted did not differ between the single-meal and the double-meal animals, apparently, the CMs produced by the two-meal model were significantly bigger. Interestingly, the RMCPII level, the mucosal mast cell activity marker, was also increased in the rats receiving two lipid meals compared with that in the rats receiving a single meal. These results suggested that the sequential lipid meal stimulated the lipid output by enhancing the secretion of larger-sized CMs into the lymph through the activation of intestinal mucosal mast cells (MMCs). The increased lymphatic lipid output and MMC activation may also contribute to postprandial lipemia. 

Using the unique lymph fistula model, we were able to monitor the real-time lipid absorption level via the small intestine without the influence of lipase in blood circulation. This model system shows four advantages over in vitro approaches, including: (1) the ability to collect hourly lymph samples continuously for 12 h after lipid infusion; (2) the animals in this model are conscious, therefore, this model is free from the complications of anesthesia that affects both CM formation and lymph flow; (3) the lymph samples that are collected and analyzed from the intestine represent the physiological response of the intestine to luminal lipids; in other words, the lipids in the lymph samples have not entered the circulation and not been metabolized by the liver or lipase in blood vessels; and (4) in our previous study [9], we found that the RMCPII concentrations in the lymph were higher compared with the serum RMCPII levels, demonstrating that the utilization of the lymph fistula rat is a more sensitive way of studying the molecules secreted by MMCs. Therefore, the conscious lymph fistula model of rats is an excellent model for monitoring the secretion of lipids from the small intestine and the inflammatory factors released by intestinal MMCs.

In this study, we first examined the lymphatic output of lipids. We found that the time to reach the peak of lipid output was significantly faster after the second lipid meal compared to the first meal, although there was no significant difference in the lymph flow rates between the two groups, which is consistent with previous reports in human studies [5,6]. In the present study, the postprandial output of all lipids, including TG, CHOL, PL, and NEFA in rat lymph reached their peaks at 2 h after the single meal, but the second peaks appeared within 1 h after the second meal. This observation suggests that pre-chylomicrons or stored lipids exist in the small intestine, which could be due to the prior lipid meal independent of dietary composition. The majority of TG accumulating in the enterocytes fuses with lipid-poor apoB-48 to form pre-chylomicrons. When the stimulus from the second meal occurs, pre-chylomicrons undergo further processing in the Golgi apparatus into mature CMs and enter the lymph, which forms a more rapid and higher peak. It seems impossible to completely process dietary fat in the intestinal lumen within such a short period. 

Previous studies have reported the existence of cytoplasmic lipid droplets (CLD) in the small intestine [8,14]. However, there was not any lipid profile in the lymph in those studies. The findings of the present study suggest that dietary lipids from the second meal might stimulate CLD mobilization and new CM formation, subsequently entering the lymph. However, the high level of lipid output induced by the second lipid meal cannot stay for a longer period. The peak started to drop at 2 h after the second lipid meal, and then remained at a lower level. This finding suggests that the stimulatory effect of the second lipid meal is temporary, which may be attributable to the limited CLD storage in the enterocytes. Both CLDs and CMs consist of a neutral lipid core surrounded by a phospholipid and cholesterol monolayer. Intestinal CLDs are dynamic: they expand in size following a fat meal, while they shrink during the postprandial state through TG lipolysis and redistribution. Abundant CLDs were detected in mice up to 12 h after an oral fat gavage [14]. Several physiological stimuli, including oral glucose from the second meal [15], glucagon-like peptide-2 [16], and sham fat feeding [17], increase the mobilization of intestinal CLDs. The physical interactions between the endoplasmic reticulum and CLD may illustrate the strong functional interactions that CLD is necessary for lipoprotein production [18]. In the present study, our results show a new finding that lipids from the second meal could trigger more lipids entering the lymph, which may be through CLD mobilization. 

Then, we examined the secretion of apolipoproteins in the lymph. ApoA-IV was significantly increased, although no significant changes could be seen in apoB-48 or apoA-I in rats with the second lipid meal. ApoA-IV, a lipid-binding protein, is synthesized and secreted by the enterocytes in response to active lipid absorption [11,12]. ApoA-IV is secreted in association with nascent CMs into the intestinal lymph. The expression of apoA-IV is highly responsive to intestinal lipid absorption and is required for CM secretion [19]. Therefore, the presence of apoA-IV in the lymph is uniquely linked to intestinal CM assembly and lipid metabolism. 

Lu et al. reported an important role for apoA-IV in facilitating the packaging of additional lipids into the CMs, and in promoting the secretion of larger CMs from the intestine into the lymph [20]. ApoA-IV is also closely located around CLDs, as confirmed by confocal microscopy [21]. It is possible that the increased apoA-IV in lymph came from pools of CLDs, which contributed to CM formation. In addition, we found that a rapid peak of apoA-IV output appeared in the lymph immediately following the second meal, which indicated that apoA-IV may correlate to the similar increasing pattern of lipid output. Since the quantity of CM remains similar between these two groups, we assume that increased CM secretion may require more apoA-IV binding. 

The intestine has been demonstrated to synthesize and secrete apoA-I [22]. However, we found that the lymphatic apoA-I level remained at comparable levels between the two groups (Figure 6a). It has been reported that CMs secreted by the enterocytes also contain apoA-I, and the latter is quickly transferred to high-density lipoprotein (HDL) in the bloodstream [23]. A previous study also found certain amounts of HDL in the lymph [24]. The unchanging apoA-I level after the second lipid meal indicated that the sequential meals had no significant impact on intestinal HDL formation. 

ApoB-48, serving as a lipid acceptor and structural protein of CMs, plays a critical role in the formation and secretion of CM particles [25]. It has been clearly demonstrated that one molecule of apoB-48 is associated with one CM particle [26]. Accordingly, the mass of lymphatic apoB-48 secretion has been used as a surrogate measure to evaluate the relative number of CM particles produced by the gut [12]. To determine whether the increased lipid transport observed in the rats receiving two lipid meals was the result of an increased number of the secreted TG-rich CM, we measured the levels of apoB-48 by Western blot. Our data showed a comparable amount of apoB-48 output in the lymph between the two-meal and one-meal rats (Figure 6c), suggesting that the secretion of apoB-48 and quantity of CM were not significantly changed by the two lipid meals. However, the second lipid meal significantly increased the TG/apoB-48 ratio (Figure 7), a marker of CM size [27], indicating the secretion of larger CMs into the lymph. This finding is different from the previous report [28], in which olive oil from the second meal increased the CM number. The reason for this difference could be due to the different lipid sources or different research subjects (humans vs. rats). However, according to our previous study [12], the continuous infusion of fat emulsion to rats did not change the apoB-48 output or CM number, instead of the increased CM size, compared with the fasting status. Therefore, the findings of the present study are consistent with our previous report [12]. 

Intestinal mucosal mast cells (MMCs) distribute predominantly in the lamina propria of the small intestine. Ji et al. [9] found that intestinal MMCs activate and degranulate to release RMCPII to the lymph during fat absorption. Once activated by stimulators, MMCs degranulate and release the preformed mediators, such as histamine and proteases, or release de novo synthesized mediators including lipid mediators, such as prostaglandin D2 and leukotrienes, as well as cytokines, such as interleukins and chemokines. There is considerable evidence showing that intestinal paracellular permeability is affected by these mast cell mediators [29,30].

During fat absorption, CMs are packaged and secreted by the enterocytes into the intercellular space by exocytosis. To enable to move the CMs from the intercellular space to the lamina propria, it requires the potential distention of the intercellular space and the possible breakage of the basement membrane. The findings from our previous studies [9] suggested that lipid infusion induced the activation of MMCs. Therefore, the role of MMC activation may be involved in the perforation of the basement membrane coupled with the loosening of the junctional complex to increase the intestinal permeability for the transport of CMs during normal processing of fat absorption. MMCs may play an important role in increasing CM secretion and transport by changing the permeability of the small intestine to adapt to the overloading of dietary lipids. 

In the present study, we found that the RMCPII level is still increased after the second lipid infusion, which is different from Ji’s report [9], in which, the RMCPII level increased quickly after the first challenge of lipids and reached the peak 1 h after the first meal, but failed to respond to the second dose of lipids [9]. This difference could be due to the following three reasons. First, the infusion location was different: the lipid meal was infused into the duodenum in Ji’s study [9], and into the stomach in our present study. Given the temporary storage and emulsion in the stomach [31], lipids may form fine droplets before entering the small intestine. Second, the duration between the two meals was different: 4 h duration in Ji’s study and 6 h duration in the present study. Under this situation, the MMCs may have time to recover sufficiently before the second lipid challenge. Third, the contents of the lipid meal were different. We used intralipid, and Ji used Liposyn II. Each has different components of oil (fatty acids), which may activate MMCs in different ways, although they both have 20% of fat. This discrepancy between Ji’s study and our current study needs to be investigated further and may provide important insight into the mechanisms of mucosal mast cell activation and active fat absorption. 

Finally, we determined the molecular mechanisms of the second lipid meal in regulating lipid absorption and transport by measuring the expression of three important genes, including *Vegfa*, *Flt1*, and *Nrp1*, in the jejunum. It is well known that lipids are absorbed mainly in the jejunum [32]. The regulation of CM uptake by lacteals is a new emerging topic. A recent study showed that VEGF-A signaling regulates CM uptake through the modulation of lacteal cell-cell junctions [33]. As a member of the VEGF family, VEGF-A plays important roles in vasculogenesis, angiogenesis, and vascular permeability. These effects are mainly mediated through binding to its receptors, VEGFR-1 (also known as *Flt1*) [34] and semaphorin receptor, NRP1 [35]. The deletion of the *Nrp1* and *Flt1* genes renders mice resistant to diet-induced obesity because of less lacteal CM uptake. Further studies have demonstrated that the absence of NRP1 and FLT1 receptors increased the VEGF-A bioavailability and signaling through VEGFR2, inducing lacteal junction zippering and CM malabsorption [33]. While we saw a trend in the increases of *Vegfa* and *Flt1* gene expression compared to those in the first lipid meal (Figure 8), the difference did not reach statistical significance, suggesting that the increased lipid transport induced by the second lipid meal is independent of the permeability of lacteals at the level of the lymphatics of the intestine [36]. This test is important because it excludes the possibility that an acute second meal affects intestinal lymphatic endothelial cells on CM absorption or secretion. Additional mechanisms, such as increased CLD mobilization or catabolism, should be considered in future studies. 

In the present study, we cannot completely rule out the possibility that a prolonged fasting time contributes to the reduced lipid output in the one-meal group compared to the output in the two-meal group. However, the different fasting periods may play a minor role in causing the differences in lipid output based on the following two reasons. First, prior to surgery, the animals were fasted overnight, plus overnight recovery; the animals were not provided with any solid food during the experimental period. Thus, the animals in both groups had been deprived of food for 36 h, before an additional 6 h fast was added to the rats in the one-meal group. Therefore, the differences in lipid output caused by the fasting period (36 h (two-meal group) vs. 42 h (one-meal group)) could be minor. Second, the lipid parameters in the fasting lymph were comparable in both groups, indicating that the fasting time did not significantly alter the basal levels. In future research, transmission electron microscopy should be applied to visualize the small intestine and lymph fluid to systematically analyze the position of lipid droplets in specific compartments and accurately quantify the size of CMs. In addition, certain intestinal transport proteins or lipid metabolic proteins should also be investigated to understand the mechanisms underlying the absorption of intestinal lipids.

## 5. Conclusions

In conclusion, we found that the second lipid meal resulted in more lipid absorption and transport, CM expansion, and mast cell activation compared to a single lipid meal. When lipids were ingested from the second meal, TG mobilization within the enterocytes was stimulated, producing larger CM particles and increasing apoA-IV output. All of these are associated with the increased RMCPII level following the second lipid infusion. The present study is important because the findings will provide more insights into the mechanisms of how the sequential lipid meal contributes to postprandial lipemia. A better understanding of the effects and underlying mechanisms induced by the second lipid meals could help to promote the development of therapeutic interventions for patients with hypertriglyceridemia and cardiovascular disease. 

## Figures and Tables

**Figure 1 genes-13-00277-f001:**
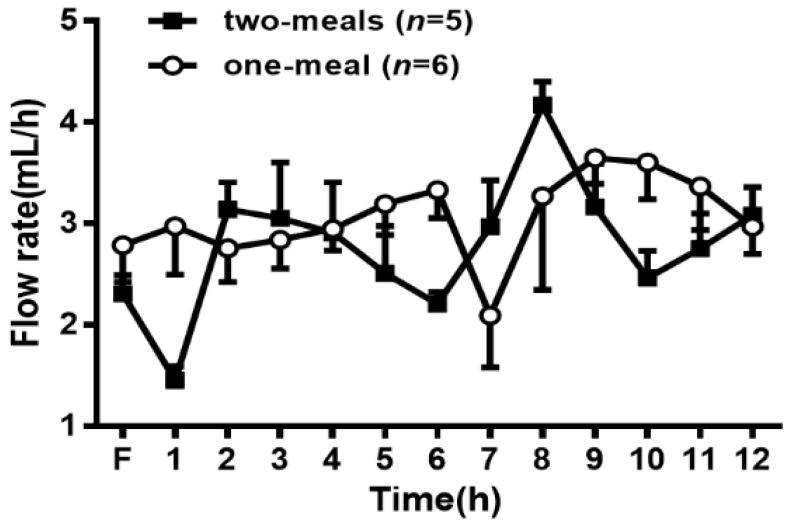
Comparison of the lymph flows rate between the two-meal and one-meal groups. Mesenteric lymph flow was measured at hourly intervals before and immediately after lipid infusion for 12 h. Values are expressed as means ± standard error of mean (SEM).

**Figure 2 genes-13-00277-f002:**
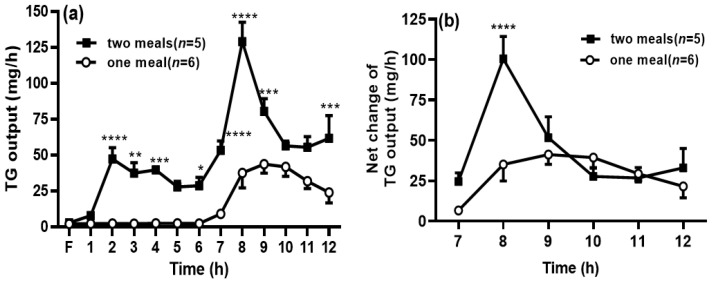
Second-meal significantly increased triglyceride (TG) transport into the lymph. The lymphatic output of the total TG mass (**a**) was measured by a TG assay kit. The net TG output was calculated by deducting the TG output at 6 h during the period within the period from 7 to 12 h (**b**). Values are expressed as means ± SEM, *n* = 5–6 rats per group. ** p* < 0.05; *** p* < 0.01; *** *p* < 0.001; and ***** p* < 0.0001, vs. the TG levels at the same time points in the one-meal group.

**Figure 3 genes-13-00277-f003:**
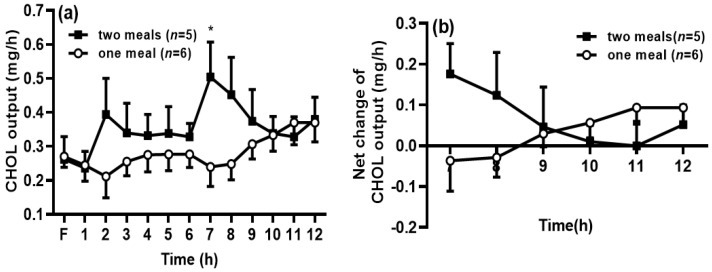
Second meal significantly increased cholesterol (CHOL) transport into the lymph. The lymphatic output of total CHOL mass (**a**) was measured by a CHOL assay kit. The net CHOL output was calculated by deducting the CHOL output at 6 h within the period from 7 to 12 h (**b**). Values are expressed as means ± SEM, *n* = 5–6 rats per group. ** p* < 0.05 compared to the CHOL level at the same time point in the one-meal group.

**Figure 4 genes-13-00277-f004:**
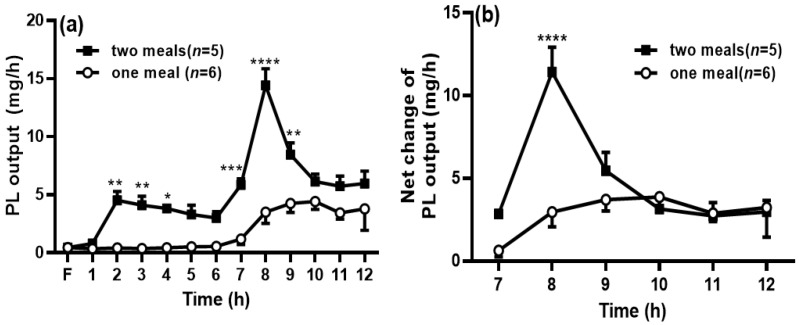
Second meal significantly increased phospholipids (PL) transport into the lymph. The lymphatic output of total PL mass (**a**) was measured by a PL assay kit. The net PL output was calculated by deducting the PL output at 6 h within the period from 7 to 12 h (**b**). Values are expressed as means ± SEM, *n* = 5–6 rats per group. ** p* < 0.05; *** p* < 0.01; **** p* < 0.001; and **** *p* < 0.0001 compared to the PL levels in the one-meal group at the same time points.

**Figure 5 genes-13-00277-f005:**
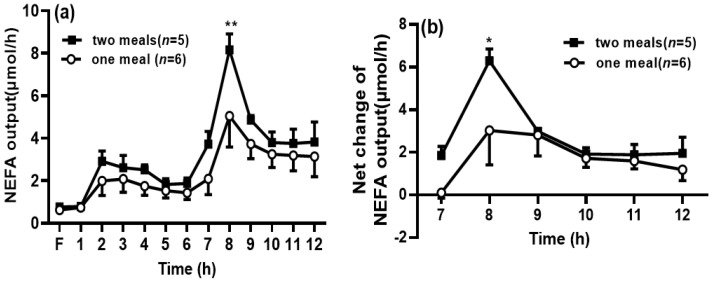
Second lipid meal significantly increased non-esterified fatty acids (NEFA) transport into the lymph. The lymphatic output of the total NEFA mass (**a**) was measured by the NEFA assay kit. The net NEFA output was calculated by deducting the NEFA output at 6 h within the period from 7 to 12 h (**b**). Values are expressed as means ± SEM, *n* = 5–6 rats per group. ** p* < 0.05 and *** p* < 0.01 compared to the NEFA levels in the one-meal group at the same time point.

**Figure 6 genes-13-00277-f006:**
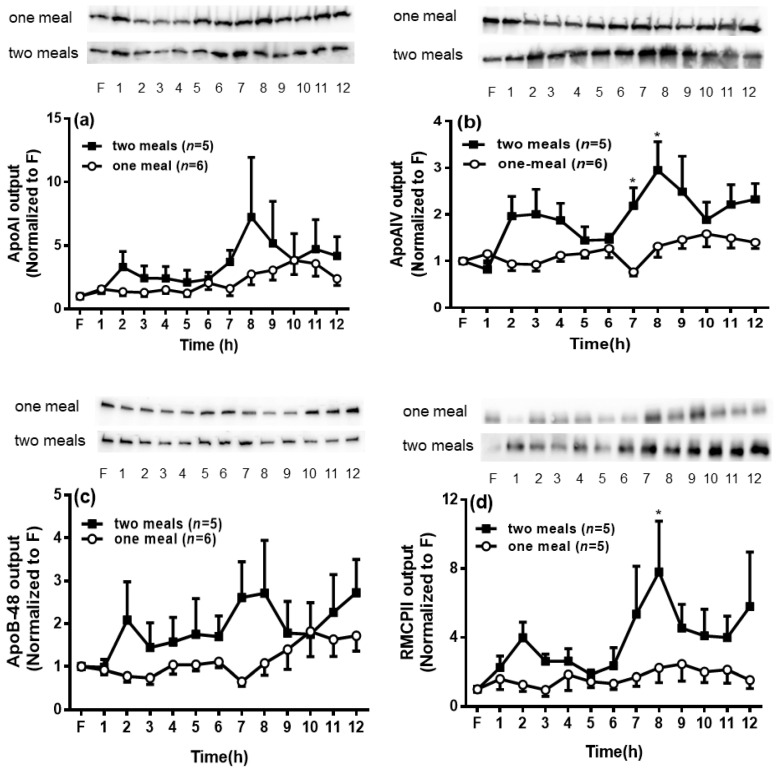
Comparison of the lymphatic outputs of apolipoproteins and rat mucosal mast cell protease II (RMCPII) between the two groups. The output ratio was calculated by densitometric quantification of the immunoblots compared with the fasting level. Apolipoprotein AI (ApoAI) output (**a**) and apolipoprotein B-48 (ApoB-48) output (**c**) remained similar level during the whole process in both groups. Apolipoprotein A-IV (ApoA-IV) output (**b**) was significantly higher at 7 h and 8 h, and RMCPII output (**d**) was higher at 8 h in the two-meal group compared to their levels in the one-meal group at the same time points. Values are expressed as means ± SEM, *n* = 5–6 rats per group. ** p* < 0.05 compared to the apoA-IV and RMCPII levels in the one-meal group at the same time point.

**Figure 7 genes-13-00277-f007:**
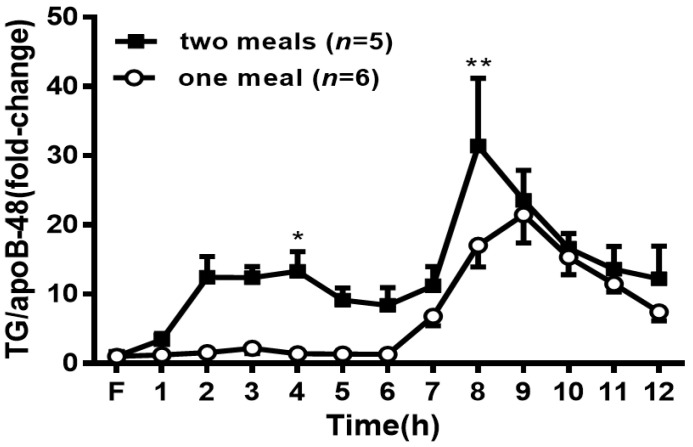
Change in the ratio of lymph TG to apoB48 during the time of treatment. Values are expressed as means ± SEM, *n* = 5–6 rats per group. ** p* < 0.05 and ** *p* < 0.01 compared to the levels in the one-meal group at the same time point.

**Figure 8 genes-13-00277-f008:**
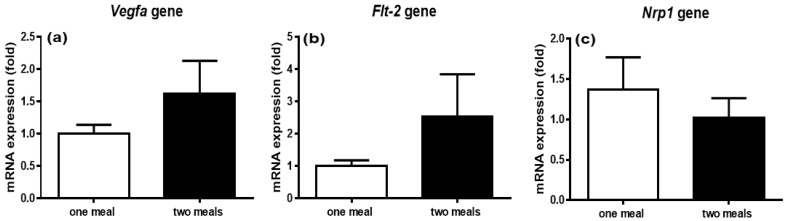
Comparison of the gene expression of vascular endothelial growth factor-A (**a**), vascular endothelial growth factor receptor 1 (**b**), and Neuropilin1 (**c**) in the jejunum. Values are expressed as means ± SEM, *n* = 4–5 rats per group. No significant differences in the mRNA levels of three genes were found between these two groups.

## Data Availability

The datasets generated and analyzed during the present study are available from the corresponding author upon reasonable request.

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
