# Peer review of "Impact of Sequential Lipid Meals on Lymphatic Lipid Absorption and Transport in Rats"

_genes, 2022, doi:10.3390/genes13020277_

Round 1

Reviewer 1 Report

The authors of this manuscript examined the impact of sequential lipid meals on lipid absorption and transport into the circulation using a lymphatic cannulation model in rats.  They report that, compared to a single high-fat meal, the increases in lymphatic lipid (TG, cholesterol, NEFA, and PL) output were significantly higher, and rises more rapidly, following a second high-fat meal. They also report that elevations of ApoA-IV and MRCPII, but not ApoB-48, were more pronounced, following a second high-fat meal, which is consistent with increased chylomicron size and lipid content in lymph. 

The study was well executed, as the authors are leading experts in using the lymph fistula model to study lipid absorption and transport. The results are clearly presented, and the authors’ interpretations seem reasonable.  The findings illustrate an effect of sequential lipid meals in lymphatic lipid transport, though the conclusion could be strengthened with additional experiments to control for potential effects of differences between one-meal and two meal groups, such as the duration of fasting.

Comments

  1. The underlying experimental design compares responses to a high-fat meal infusion either: A) following a 12 h fast or B) following a high-fat meal and a 6-h fast. As such, it is impossible to determine if the differences observed are due to the previous exposure to a high-fat meal or to the reduced fasting period. 
  2. The level of NEFA increased after the 1st hour in the one meal group, while the mice were still fasted. Why?
  3. ApoAIV increased after the initial lipid infusion in the two-meal group, but not in the one meal group. Why?
  4. Line 70. Please describe diet of rats prior to surgery.
  5. Line 84. Please clarify if “fasting” lymph collection after overnight recovery was made during saline/glucose infusion.
  6. Figures 2b, 3b, 4b, and 5b. For Net change, please consider including 6th h time point in each graph to serve as a baseline.  Were any of the AUCs significantly different in any of the graphs?
  7. Line 239. Change “contains on” to “contains one”.
  8. Line 267-268. Consider clarifying that the two-meal group is being compared to the one-meal shortened fast
  9. Line 300. Consider acknowledging that the stored lipid in enterocytes could be due to the recent meal independent of dietary composition.
  10. Line 396. Change “between Ji’ study” to “between Ji’s study”.

Author Response

Responses to the comments

Review report 1

  1. The underlying experimental design compares responses to a high-fat meal infusion either: A) following a 12 h fast or B) following a high-fat meal and a 6-h fast. As such, it is impossible to determine if the differences observed are due to the previous exposure to a high-fat meal or to the reduced fasting period.

A:  The reviewer is right. In this study, we cannot completely rule out the possibility that prolonged fasting time contributes to the reduced lipid output in one-meal group, compared to the output in two-meal group. However, we think that the different fasting periods may play a minor role in causing such differences based on following two reasons. First, prior to surgery, the animals were fasted overnight, plus overnight recovery, the animals did not consume solid food during the experimental period, although they were infused via the gastric tube with a saline solution containing 5% glucose.  Thus, the animals in both groups had been deprived of food for 36 h, before an additional 6 h fast was added to the rats in one-meal group. Therefore, the differences in lipid output caused by fasting period [36 h (two-meal group) vs. 42 h (one-meal group] could be trivial.  Second, the lipid parameters in the fasting lymph were comparable between both groups, indicating that the fasting time did not significantly alter the basal levels.  We want to thank the reviewer for the thoughtful comment, and we have revised the manuscript by addressing this possibility in Discussion section (please see Lines 424-434).

2.The level of NEFA increased after the 1st hour in the one meal group, while the mice were still fasted. Why?

A:  Thank you for pointing this out.  In this study, we used NEFA output to compare the difference between one-meal and two-meal groups.  The NEFA output is NEFA concentration multiplied by lymph flow rate. Since lymph flow rate varied in different animals, the increased NEFA output was due to increased lymph flow rate in one-meal group.  Importantly, compared to the NEFA levels at fasting, the NEFA levels at the 1st hour in the one-meal group were slightly increased, and the difference did not reach statistical significance.

3.ApoAIV increased after the initial lipid infusion in the two-meal group, but not in the one meal group. Why?

A: Thank you for the comment.  While there was a trend of increase in apoAIV level after the infusion of the 1st lipid meal in two-meal groups, the difference in the apoAIV outputs was not significant statistically, compared to the apoAIV level either at the same time period (1 h to 6 h) or the time period after lipid infusion (7 h to 12 h) in one-meal group.  These data indicated the variation in response to lipid meal in different animals.

4.Line 70. Please describe diet of rats prior to surgery.

A: Thank you for the comment.  We have included the diet information in the revised manuscript (please see Lines 70-71).

5.Line 84. Please clarify if “fasting” lymph collection after overnight recovery was made during saline/glucose infusion.

A: The sentence has been revised in line 86, thanks.

6.Figures 2b, 3b, 4b, and 5b. For Net change, please consider including 6th h time point in each graph to serve as a baseline.  Were any of the AUCs significantly different in any of the graphs?

A: Thank you for the comment. Since the net changes in the period of 7 h to 12 h were obtained after subtracting the value at 6 h in each animal, all numbers at the 6th h were treated as 0.  It may not be necessary to add the “0” at 6 h in these figures.  According to the reviewer’s suggestion, we have re-calculated all the AUCs, and found only the AUCs in PL output was significantly increased in two-meal group than that in one-meal group. In order to keep the four figures consistent (i.e., Figures 2, 3, 4, and 5), we did not include these AUC data in the revised manuscript, instead we have presented the AUC data in words (please see lines 202-204).

7.Line 239. Change “contains on” to “contains one”.

A:  Thank you for pointing it out. This word has been changed (line 242).

8.Line 267-268. Consider clarifying that the two-meal group is being compared to the one-meal shortened fast

A: We want to thank the reviewer, and this sentence has been revised (please see line 271).

9.Line 300. Consider acknowledging that the stored lipid in enterocytes could be due to the recent meal independent of dietary composition.

A: The sentence has been revised in line 304-305, thanks.

10.Line 396. Change “between Ji’ study” to “between Ji’s study”.

A:  Thank you for pointing it out.  The word has been changed in line 401.

Reviewer 2 Report

The work presented by Qi Zhu et al. focused on how a two-meal pattern affects the absorption and lymphatic transport of lipids in adult rats. Data sound and are interesting in the understanding of lipid absorption.

However, some points must be considered :

  • The lipid infusion composition and caloric content should be described.
  • From the metabolic point of view, it will be interesting to have additional plasma parameters such as glycemia and insulinemia in rats after one or two meals.
  • How the authors deal with the standardization of values obtained from the two groups (one-meal and two-meals)?

Author Response

Review report 2

The work presented by Qi Zhu et al. focused on how a two-meal pattern affects the absorption and lymphatic transport of lipids in adult rats. Data sound and are interesting in the understanding of lipid absorption.

However, some points must be considered:

The lipid infusion composition and caloric content should be described.

A:  Thank you for the comment. The composition of intralipid is 20% soybean oil, 1.2% egg yolk phospholipids, 2.25% glycerin and water. The Intralipid caloric value is 2.0 kcal per ml.  This information has been included in the revised manuscript (line 90-91).

From the metabolic point of view, it will be interesting to have additional plasma parameters such as glycemia and insulinemia in rats after one or two meals.

A:  We want to thank you for the insightful comment. Since we mainly focused on the lipid absorption and metabolism in the small intestine, we did not measure the glucose level in plasma. In addition, we infused 5% glucose/saline to rats during the entire experimental period, which may not be a good animal model to monitor the glycemia and insulinemia. However, we will keep this important comment in our mind, and will address it in our future studies.

How the authors deal with the standardization of values obtained from the two groups (one-meal and two-meals)?

A:  We want to thank the reviewer for the comment.  When we measured lipid concentration with commercial kits, standard substance control and negative control were always included in each assay.

For lymphatic output of apolipoproteins and RMCPII determined by Western blot analysis, all antibodies were used and validated in our previous publications. (Gastroenterology, 2016 Nov;151:923-932, PMID: 27436071, and Am J Physiol Gastrointest Liver Physiol. 2012;302:G1292-300, PMID: 22461027)

For q-PCR test, we measured the gene expression by applying the housekeeping gene β-actin as the internal control to normalize the mRNA levels.  The TaqMan probes, purchased from ThermoFisher Scientific, were used in our recent publication (J Physiol. 2021;599:5015-5030. PMID: 34648185)

Round 2

Reviewer 2 Report

Authors have answered most of my concerns.